# Characteristics of Mandibular Arch Forms in Patients with Skeletal Mandibular Prognathism

**DOI:** 10.3390/diagnostics13203237

**Published:** 2023-10-17

**Authors:** Erika Ichikawa, Chie Tachiki, Kunihiko Nojima, Satoru Matsunaga, Keisuke Sugahara, Akira Watanabe, Norio Kasahara, Yasushi Nishii

**Affiliations:** 1Department of Orthodontics, Tokyo Dental College, Chiyoda, Tokyo 101-0061, Japan; ichikawaerika@tdc.ac.jp (E.I.); nishii@tdc.ac.jp (Y.N.); 2Fujiseki Dental Clinic, Chiyoda, Tokyo 101-0041, Japan; 3Department of Anatomy, Tokyo Dental College, Chiyoda, Tokyo 101-0061, Japan; matsuna@tdc.ac.jp; 4Department of Oral Pathobiological Science and Surgery, Tokyo Dental College, Chiyoda, Tokyo 101-0061, Japan; ksugahara@tdc.ac.jp; 5Department of Oral & Maxillofacial Surgery, Tokyo Dental College, Chiyoda, Tokyo 101-0061, Japan; akirawat@tdc.ac.jp; 6Department of Histology and Developmental Biology, Tokyo Dental College, Chiyoda, Tokyo 101-0061, Japan; nkasahara@tdc.ac.jp

**Keywords:** arch form, skeletal mandibular prognathism, surgical orthodontics, facial pattern

## Abstract

Arch forms in orthodontics are considered to affect occlusal stability. This study’s subjects were 47 patients (Class III S group) who visited the Chiba Dental Center of Tokyo Dental College and were surgical orthodontic cases, and 60 patients with Class I malocclusion were selected as the control group. A mandibular model of each subject was plotted with each tooth on a digitizer. The clinical bracket points of each tooth were plotted, and intercanine and intermolar measurements were taken. The least squares method was used to fit a quartic equation, and the arch form was drawn. The Class IIIS group was divided by Wits appraisal and facial pattern into a dolichofacial or brachyfacial pattern, and arch forms were compared. The results show that the Class IIIS group had a significantly smaller intermolar width, canine depth, and molar depth and a significantly larger canine W/D ratio. In those with a dolichofacial pattern, the anterior curve of the arch form tended to be flat and the posterior curve narrower. This is because, in skeletal mandibular prognathism, the mandibular anterior shows lingual tipping, and the molars show palatal tipping due to dental compensation, and it was inferred that this tendency was higher in high-angle cases.

## 1. Introduction

Surgical orthodontic treatment can be applied to severe skeletal mandibular prognathism to improve the three-dimensional disharmony of the upper and lower jaws as well as construct a functional occlusion [1]. When using cephalometric X-rays, dental models, and computed tomography (CT), it is necessary to set highly accurate goals for preoperative orthodontic treatment, such as harmonization of the maxillary and mandibular dental arch forms, optimization of the dental axis inclination of the anterior teeth for each jaw (dental decompensation), and prediction of the direction and amount of movement of the maxillary and mandibular bones. In addition, it is now possible to accurately simulate postoperative facial changes, and morphological restoration has evolved even further than before. However, from the perspective of maintaining and stabilizing occlusion, many issues, such as postoperative relapse, remain.

In orthodontic treatment, there is a consensus that the determination of a patient’s post-treatment arch form not only satisfies esthetic requirements but also contributes to occlusal stability by maintaining the original mandibular intercanine width to prevent relapse and keep the original arch form unchanged [2]. Clinically, it is considered useful to determine the arch form for a pre-treatment mandibular dental model. Therefore, many studies have been conducted on arch form differences by ethnicity [3,4,5,6,7] and malocclusion type [8,9,10,11] in orthodontically treated patients. On the other hand, there are few reports of arch forms of cases with skeletal deviation. Since such cases are often treated clinically, it is meaningful to investigate these issues. Therefore, the present study was designed to characterize the arch form of the mandible in skeletal mandibular prognathism.

## 2. Materials and Methods

### 2.1. Subjects

The subjects were 47 patients with skeletal mandibular prognathism (Class IIIS group) who visited the Chiba Dental Center of Tokyo Dental College between 2015 and 2023. The inclusion criteria were diagnosis of mandibular prognathism requiring surgical orthodontic treatment and an arch length discrepancy of 4 mm or less. The exclusion criteria were abnormal tooth morphology or missing teeth, a menton (the extreme inferior point of the chin) deviation of 4 mm or more, restorations extending to the contact point, cusp tips and incisal edges, and a history of orthodontic treatment or congenital disease. Two individuals with more than 10 years of experience in orthodontic treatment were involved in the diagnoses. Cephalometric radiographs of the subjects were analyzed to measure the ANB (an item evaluating the anteroposterior position of the maxillary and mandibular alveolar bases) angle and perform Wits appraisal (an item evaluating the anteroposterior relationship of the maxilla and mandible) to assess anteroposterior skeletal positioning. The facial pattern was comprehensively determined from the results of Ricketts’ analysis of the facial axis, mandibular plane angle, lower facial height, mandibular arc, and total facial height, which were used to determine the vertical position of the mandible. For each measurement, ±1 SD from the standard value was divided into three parts, with −0.5 for 1/3 SD in the dolicho tendency, 0 for 1/3 SD in the center, and +0.5 for 1/3 SD in the brachy tendency. More than ±1 SD but less than ±2 SD was set to ±1, more than ±2 SD but less than ±3 SD was set to ±1.5, more than ±3 SD but less than ±5 SD was set to ±2.5, and more than ±4 SD was set to ±3. The cumulative total for each item was classified as follows: −3.0 or less was classified as a dolichofacial pattern (high-angle cases), −2.5 to +2.5 as a mesiofacial pattern, and +3 or more as a brachyfacial pattern (low-angle cases). The subjects’ characteristics are shown in Table 1.

The control group consisted of 60 patients (27 males and 33 females; mean age 16.3 years) diagnosed with Class I malocclusion who underwent orthodontic treatment at the same institution. The inclusion criteria were molar relationship Class I, skeletal Class I, and a menton deviation within 4 mm. Pre-treatment mandibular dental arch models of these subjects were used.

### 2.2. Measurement Method

The occlusal planes of the mandibular models were photocopied with a ruler included for magnification correction. The photocopied images were placed in a digitizer (DigitizerII, Wacom, Saitama, Japan), and most facial portions of 13 proximal contact areas around the arch were digitized and set as the coordinate origin between the central incisors (point 7). A line (Line A) connecting the contact points of the first and second premolars (point 3 and point 11) on both sides was drawn, and another line (Line B) connecting the contact points of the second premolar and first molar (point 2 and point 12) was also drawn. The X-axis was set to be parallel to the average of the slopes of these two lines, and the perpendicular line to it was the Y-axis (Figure 1). Twelve clinical bracket points corresponding to the bracket and tube attachment position of each tooth were set as representative points for each tooth [3]. A line connecting the mesial and distal contact points of each tooth (embrasure line) was drawn on the coordinates. For incisors, canines, and bicuspids, a line perpendicular to the embrasure line was drawn from the middle of the embrasure line, and for molars, a line perpendicular to the embrasure line was drawn from the proximal 1/3 of the line to the buccal side of each tooth (Figure 2). On this line, a clinical bracket point was set using the mandibular tooth thickness data [12].

All contact points were plotted by a single measurer. The measurement error was assessed by statistically analyzing the differences between duplicate measurements taken at least 2 weeks apart on 24 points selected at random. The measurement error was less than 5% of the average value of the measurement and within acceptable limits.

### 2.3. Measurement Items (Figure 3)

The measurement items included were as follows:Intercanine width, the distance between the canine clinical bracket points;Intermolar width, the distance between the first molar clinical bracket points;Canine depth, the shortest distance from a line connecting the canine clinical bracket points to the origin between the central incisors;Molar depth, the shortest distance from a line connecting the first molar clinical bracket points to the origin between the central incisors;Canine W/D ratio, the ratio of the intercanine width to the canine depth;Molar W/D ratio, the ratio of the intermolar width to the molar depth.

The Class IIIS group and control group were compared for the above items.

**Figure 3 diagnostics-13-03237-f003:**
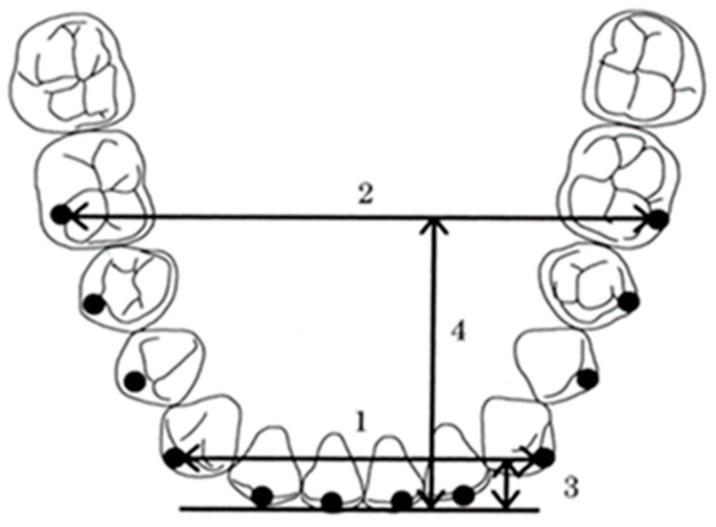
Twelve clinical bracket points and four linear measurements of arch dimensions. 1: intercanine width; 2: intermolar width; 3: canine depth; 4: molar depth.

To examine the effect of the degree of skeletal mandibular prognathism on the arch form, the Class IIIS group was further subdivided into two subgroups—Wits appraisal values of −10 mm or more and −10 mm or less—and the canine W/D ratios and molar W/D ratios were compared between the two subgroups. In addition, a similar comparison between the two subgroups was made for those classified as a brachyfacial pattern (low-angle cases) and dolichofacial pattern (high-angle cases) to examine the effect of facial patterns on arch forms.

### 2.4. Arch form Comparison

The coordinates of the twelve clinical bracket points were set, and the quartic equation (y = ax^4^ + bx^3^ + cx^2^ + e) was fit using the least squares method. The average arch form with clinical bracket points in the Class IIIS group and the average mandibular arch form in the control group were drawn on the same XY coordinates of reference and compared. Arch forms were drawn for each facial pattern and compared with the control group.

In addition, each dental arch form was selected from the square, ovoid, and tapered arch forms (OrthoForm, 3M Unitek, Monrovia, CA, USA) according to the criteria of McLaughlin et al. [13]. The arch form that best matched that which best fit the 8 clinical bracket points from the first premolar to the first premolar was selected (Figure 4).

### 2.5. Statistical Analysis

The sample size was calculated using the mean and standard deviation of the canine W/D ratio for each group in our preliminary study, with an alpha error of 5% and a beta error of 5%, and 29 patients were required in each group.

The result of the Shapiro–Wilk test in this study was *p* > 0.05, indicating normality. An unresponsive Student’s *t*-test was used to compare the two groups, and a chi-square test was used to examine the arch forms’ frequency of occurrence. Significance was set at *p* < 0.05. All statistical analyses were performed using Microsoft Office Excel (version 2306, Microsoft, Redmond, WA, USA).

## 3. Results

Compared with the control group, the Class IIIS group showed no significant difference in intercanine width, but the intermolar width, canine depth, and molar depth were significantly smaller, and the canine W/D ratio was significantly larger (Table 2).

Table 3 shows the canine W/D ratios, and the molar W/D ratios were not significantly different between the Wits appraisal of −10 mm or more and −10 mm or less subgroups.

In the facial pattern comparison, both the canine W/D ratio and the molar W/D ratio were significantly smaller in the brachyfacial pattern.

Comparing the Class IIIS and control groups’ arch form averages using the quartic equation showed that the Class IIIS group had a wider arch anteriorly and a narrower arch posteriorly. A comparison of facial patterns showed that the brachyfacial pattern was narrower anterior and posterior to the arch, whereas the dolichofacial pattern was wider anterior to the arch and narrower posterior to the arch (Figure 5). The frequency of archwire forms in each group is also shown. In the Class IIIS group, 42.5% were square, 31.9% were ovoid, and 25.6% were tapered; in the Class I group, 53.3% were square, 38.3% were ovoid, and 8.3% were tapered. The chi-square test showed a statistically significant difference between the two groups (Table 4).

## 4. Discussion

Based on research on relapse, Little recommended that pre-treatment dental arch morphology should be used as an important indicator for post-treatment dental arch morphology [14]. Many studies have shown that maintaining the pre-treatment intercanine width and arch morphology of the mandibular dentition after treatment is beneficial for long-term occlusal stability [15,16].

However, with the recent development of elastic-rich wire materials and the pre-adjusted appliance system, various companies have introduced ready-made archwire forms based on the average arch form obtained from normal occlusion. Therefore, those wire materials have been increasingly used in full-scale orthodontic treatment, mainly in the leveling and alignment treatment stage. Because of its characteristics, it is difficult for clinicians to adjust the arch form for each patient. Therefore, in many cases, it is considered appropriate to select from several ready-made archwire forms that approximate the pre-treatment arch form, taking into account race and the type of malocclusion, and to make fine adjustments for each patient.

In research on arch forms in Japanese people, Sebata et al. [17] and Kosaka et al. [18] reported on Class I malocclusions, and Kayukawa et al. [8] and Imaizumi et al. [9] reported on patients with Class III malocclusions. They stated that there is no significant difference in arch width between those with Class III malocclusions and those with Class I malocclusions, but the arch length is significantly greater. However, there are no reports of arch forms limited to patients with skeletal mandibular prognathism diagnosed as requiring surgical orthodontic treatment. As surgical orthodontic treatment is a major part of orthodontic treatment [19], it is important to compare and contrast them.

For the measurement method in the present study, we referred to the method of Nojima et al. [3], who compared the arch forms of Caucasian and Japanese people using an arch form with a series of clinical bracket points, rather than an arch form with a cusp and incisor series, to obtain guidelines that can be clinically applied.

The results of the present study showed that mandibular prognathism was not significantly different from Class I malocclusion in terms of intercanine width, and it showed significantly smaller values in intermolar width, canine depth, and molar depth. Therefore, the canine W/D ratio was significantly greater, and the molar W/D ratio was not significantly different.

Nojima et al. [3] and Braun et al. [20] stated that the intercanine and intermolar widths were significantly greater in Class III malocclusions than in Class I malocclusions. In terms of arch length, Kayukawa et al. [8] stated that both canine and molar depths were significantly larger. Thus, it can be inferred that skeletal factors affected the results that differed from those of the present study. In other words, since the anteroposterior occlusal relationship in skeletal mandibular prognathism presents a significant Class III molar relationship, many of the mandibular first molars are in occlusion with the maxillary premolars, which are in occlusion with a more constricted area, resulting in lingual tipping of the mandibular molars and a smaller intermolar width. In addition, the arch length is small due to dental compensation in the mandibular anterior.

When comparing arch forms, skeletal mandibular prognathism showed flatter curves in the anterior arch and narrower molars compared with Class I malocclusion. In the present study, we assumed that the more pronounced skeletal Class III is, the stronger this tendency becomes, and we used Wits appraisal for comparison. ANB and Wits appraisal are the principal cephalometric parameters that indicate the anteroposterior position of the skeleton. However, in skeletal mandibular prognathism, the Wits appraisal, which was used in the present study, is considered to be more indicative of anteroposterior deviation [21]. However, this comparison showed no significant differences, suggesting that anteroposterior deviation was not the only effect. In addition, a comparison of the vertical position of the skeleton in terms of the facial pattern showed significant differences, indicating that the arch form differs depending on the vertical factor. In the brachyfacial pattern (low-angle cases), there was little anterior curve flattening, whereas in the dolichofacial (high-angle cases) pattern, anterior curve flattening, a characteristic of skeletal Class III, was high. This may be because of the longer mandibular length in high-angle cases, resulting in higher dental compensation in the anterior. Since this vertical deviation is not manifested numerically as an anteroposterior deviation of the skeleton, it is thought to have had little effect on the Wits appraisal.

Engel [22] classified archwire forms into nine categories, and Raberin et al. [23] classified them into five categories, trying to accommodate clinically various arch forms. Moreover, Nojima et al. [3] classified mandibular arch forms into square, ovoid, and tapered according to McLaughlin et al.’s criteria [13] and characterized their arch forms based on their frequency of occurrence, such that no statistically significant differences were found between races in each arch form. Therefore, this classification into three categories was considered appropriate in orthodontic practice, and this classification method was also used in this study.

The frequency of occurrence of ready-made archwire forms in this study was almost half for the square form in the Class IIIS group, with the ovoid and tapered forms each accounting for half of that number. In the Class I group, square accounted for more than half, ovoid for approximately one-third, and tapered for a small number, indicating that the frequency of occurrence differed between the two groups. Namely, the frequency of occurrence of the square form was similar, but the Class III S group had an increased rate of the tapered form compared with the Class I group. This is because the Class III S group in this study consisted of 24 brachy, 13 mesio, and 10 dolicho cases, with a large proportion of brachy cases, which may have increased the number of tapered cases. The results of this study indicate that the arch form of skeletal mandibular prognathism is often square and that vertical skeletal relationships, such as facial patterns, may also have a significant effect on the arch form.

According to the results of this study, skeletal mandibular prognathism showed lingual tipping of the mandibular anterior due to dental compensation and palatal tipping of the molars also due to dental compensation [24]. Compared with Class I malocclusion, there was a difference in flattened curves in the anterior and a narrowed arch form in the molars. In presurgical orthodontics, it is considered important to release these dental compensations and to set appropriate tooth axes and arch forms about the jawbone. Therefore, it is thought that a specific archwire form should not be applied to all mandibular prognathism patients. However, no difference was observed in the intercanine width compared with Class I malocclusion, suggesting that the preoperative intercanine width should be used as a reference for arch form selection.

Little et al. [14] focused on the mandibular arch form from the perspective of relapse, so the maxillary arch form was not investigated. However, the maxillary arch form of patients with mandibular prognathism may also be affected and should be investigated in the future as well [25].

In addition, patients with skeletal mandibular prognathism often have facial asymmetry, which may affect the arch form. Therefore, patients with facial asymmetry were excluded from this study.

Facial and intraoral 3D scanning has become widely used in recent years [26], and applying these techniques for facial soft tissue analysis and tooth axis analysis will contribute to more refined surgical orthodontic treatment in the future.

## 5. Conclusions

In this study, the mandibular arch form of skeletal mandibular prognathism for which surgical orthodontics was indicated was compared with that of Class I malocclusion, and its characteristics were clarified.

First, compared with Class I malocclusion, the mandibular arch form of skeletal mandibular prognathism showed no significant difference in intercanine width; a significantly smaller intermolar width, canine depth, and molar depth; and a significantly larger intercanine W/D ratio.

Second, arch forms drawn using a quartic equation demonstrated that skeletal mandibular prognathism showed flatter curves in the anterior and narrower widths in the molars compared with Class I malocclusion. This is presumably because the anterior and molars show dental compensation due to the incongruity of the position of the maxilla and mandible.

Finally, a comparison of arch forms by facial pattern showed that the anterior of the dolichofacial pattern (high-angle cases) was nearly flat, indicating higher dental compensation in the anterior.

## Figures and Tables

**Figure 1 diagnostics-13-03237-f001:**
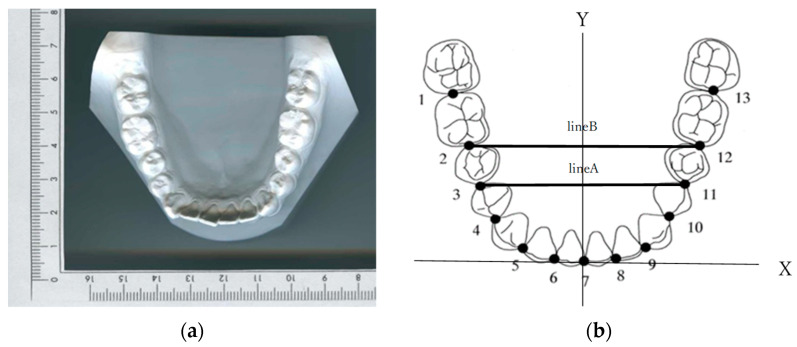
How to set the coordinate axes in the dental models. (**a**) The occlusal planes of the mandibular models were photocopied with a ruler included for magnification correction. (**b**) The photocopied images were placed in a digitizer, and most facial portions of 13 proximal contact areas around the arch were digitized and set as the coordinate origin between the central incisors (point 7). Line A: line connecting point 3 and point 11; Line B: line connecting point 2 and point 12; X-axis: adjusted to be parallel to the average of the slopes of Line A and Line B; Y-axis: adjusted to be perpendicular to the X-axis.

**Figure 2 diagnostics-13-03237-f002:**
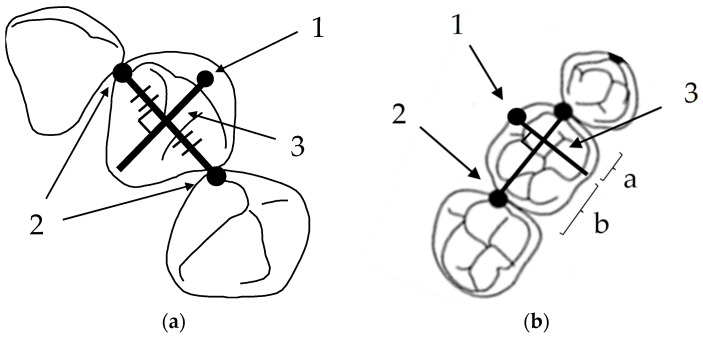
How to set the clinical bracket points for each tooth. (**a**) The clinical bracket points of anterior teeth. 1: The clinical bracket points; 2: contact points; 3: perpendicular to the midpoint of the line connecting the contact points. (**b**) The clinical bracket points of premolars and molars. 1: The clinical bracket points; 2: contact points; 3: perpendicular to the mesial third of the line connecting the contact points (a:b = 1:3).

**Figure 4 diagnostics-13-03237-f004:**
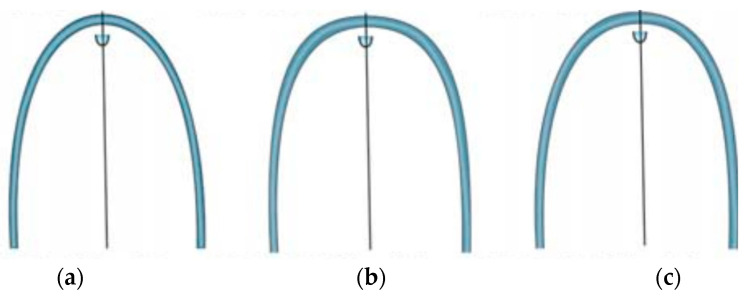
Tapered, square, and ovoid arch forms used for classification (OrthoForm, 3M Unitek, Monrovia, CA, USA): (**a**) tapered; (**b**) square; and (**c**) ovoid.

**Figure 5 diagnostics-13-03237-f005:**
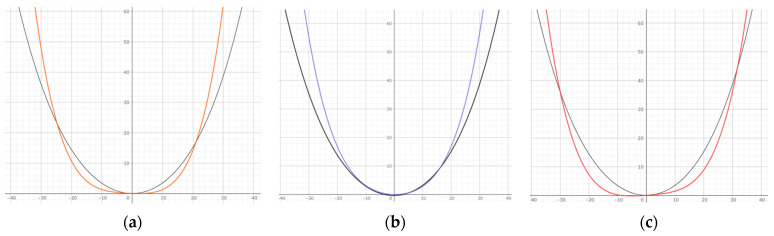
Arch form comparison. Black: control group; orange: average of Class IIIS group; blue: brachy in Class IIIS group; red: dolicho in Class IIIS group. (**a**) Comparison of averages of Class IIIS group and control group; (**b**) comparison of brachy in Class IIIS group and control group; (**c**) comparison of dolicho in Class IIIS group and control group.

**Table 1 diagnostics-13-03237-t001:** Patients‘ characteristics (*n* = 47).

Sex	Male: 18		
	Female: 29		
Age (year)	24.3 ± 9.7		
ANB (°)	−2.5 ± 2.2	Max.: 0	Min.: −5.5
Wits appraisal (mm)	−12.6 ± 4.5	Max.: −6	Min.: −25
Facial pattern	Brachy: 24	Mesio: 13	Dolicho: 10

**Table 2 diagnostics-13-03237-t002:** Comparison of variables between Class IIIS group and control group.

Groups	Class IIIS Group		Control Group		
Measurement Items	Average	SD	Average	SD	*p*-Value
Intercanine width (mm)	29.17	1.87	29.89	1.52	NS
Intermolar width (mm)	48.69	3.11	50.70	2.81	0.001
Canine depth (mm)	4.64	1.14	5.66	1.01	0.000
Molar depth (mm)	24.91	2.15	26.28	1.94	0.000
Canine W/D ratio	6.73	1.94	5.43	0.87	0.000
Molar W/D ratio	1.97	0.21	1.94	0.15	NS

NS: not significantly different.

**Table 3 diagnostics-13-03237-t003:** Comparison of arch forms by Wits and facial pattern in Class IIIS group.

Groups	Wits ≤ −10		Wits > −10			Brachy		Dolicho		
Measurement Items	Average	SD	Average	SD	*p*-Value	Average	SD	Average	SD	*p*-Value
Canine W/D ratio	7.71	3.77	6.47	2.22	NS	6.08	1.10	8.28	3.03	0.004
Molar W/D ratio	1.97	0.18	1.99	0.25	NS	1.91	0.16	2.07	0.22	0.020

NS: not significantly different.

**Table 4 diagnostics-13-03237-t004:** Comparison of frequency distributions of square, ovoid, and tapered arch forms between Class IIIS group and control group.

Groups	Square	Ovoid	Tapered	*p*-Value
Class IIIS group	20 (42.5%)	15 (31.9%)	12 (25.6%)	0.016
Control group	32 (53.3%)	23 (38.3%)	5 (8.3%)	

## Data Availability

The data presented in this study are available upon request from the corresponding author.

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
