# Peer review of "Characteristics of Mandibular Arch Forms in Patients with Skeletal Mandibular Prognathism"

_diagnostics, 2023, doi:10.3390/diagnostics13203237_

Round 1
Reviewer 1 Report
Dear Authors,
you made a great work! However, some improvements are suggested before acceptance.

Reviewer 2 Report
Dear Authors,
I’ve extensively read the manuscript titled “Characteristics of Mandibular Arch Forms in Patients with 2 Skeletal Mandibular Prognathism. The aim of this study was to characterize the arch form of the mandible in skel-53 etal mandibular prognathism. The methodology is appropriate and quite linear with recent evidences/ studies on this topic. In particular, it is not easy to perform good systematic review on 3d printing in orthodontics due to the extremely high confounding variables (technology, resins etc) but the authors made a good job. I’ve not major concerns in this regard.
Some aspects must be addressed before considering the manuscript suitable for publication:
- A revision of scientific English language is required.
- Arch form could be significantly influenced by the presence of mandibular shift which determines asymmetry of the alveolar process on the upper arch. There are recent significant contributions in the literature about this aspect and should be cited
https://pubmed.ncbi.nlm.nih.gov/29474543/
- Also authors should address in the discussion the implication pf the above mentioned findings in the contest of their study design that has been focused on the mandibular arch
.-authors should argue about the possible distinction between the subjects included in their examination and those with mild or mixed form of class III malocclusion
- In general, the discussion section should be improved, even considering the above mentioned topics
requires significant revision
Round 2
Reviewer 2 Report
the authors have successfully satisfied all my previous concerns. in my opinion, the manuscript can be published